# Structure-Based Bioisosterism Design, Synthesis, Biological Evaluation and In Silico Studies of Benzamide Analogs as Potential Anthelmintics

**DOI:** 10.3390/molecules27092659

**Published:** 2022-04-20

**Authors:** Franco Vairoletti, Margot Paulino, Graciela Mahler, Gustavo Salinas, Cecilia Saiz

**Affiliations:** 1Laboratorio de Química Farmacéutica, Departamento de Química Orgánica, Facultad de Química, Universidad de la República, Montevideo 11800, Uruguay; fvairoletti@fq.edu.uy (F.V.); gmahler@fq.edu.uy (G.M.); 2Graduate Program in Chemistry, Facultad de Química, Universidad de la República, Montevideo 11800, Uruguay; 3LaBioFarMol, Departamento de Experimentación y Teoría de la Estructura de la Materia y sus Aplicaciones, Facultad de Química, Universidad de la República, Montevideo 11800, Uruguay; margot@fq.edu.uy; 4Departamento de Biociencias, Facultad de Química, Universidad de la República, Montevideo 11800, Uruguay; 5Laboratorio de Biología de Gusanos, Institut Pasteur de Montevideo, Montevideo 11400, Uruguay

**Keywords:** bioisosterism, nematicide, *ortho*-substituted benzamide, *C. elegans*, *E. granulosus*

## Abstract

A recent screen of 67,012 compounds identified a new family of compounds with excellent nematicidal activity: the *ortho*-substituted benzamide families Wact-11 and Wact-12. These compounds are active against *Caenorhabditis elegans* and parasitic nematodes by selectively inhibiting nematode complex II, and they display low toxicity in mammalian cells and vertebrate organisms. Although a big number of benzamides were tested against *C. elegans* in high-throughput screens, bioisosteres of the amide moiety were not represented in the chemical space examined. We thus identified an opportunity for the design, synthesis and evaluation of novel compounds, using bioisosteric replacements of the amide group present in benzamides. The compound Wact-11 was used as the reference scaffold to prepare a set of bioisosteres to be evaluated against *C. elegans*. Eight types of amide replacement were selected, including ester, thioamide, selenoamide, sulfonamide, alkyl thio- and oxo-amides, urea and triazole. The results allowed us to perform a structure–activity relationship, highlighting the relevance of the amide group for nematicide activity. Experimental evidence was complemented with in silico structural studies over a *C. elegans* complex II model as a molecular target of benzamides. Importantly, compound Wact-11 was active against the flatworm *Echinococcus granulosus*, suggesting a previously unreported pan-anthelmintic potential for benzamides.

## 1. Introduction

Helminths are parasites that affect humans, animals and plants. Nematode and platyhelminths infections in humans are responsible for significant morbidity and mortality worldwide, particularly in developing countries, while infections in livestock and crops cause a huge economic burden, with losses estimated at billions of USD per year [1]. The main control strategy for helminth infection relies on the use of anthelmintic drugs such as benzimidazoles, macrocyclic lactones, imidothiazoles, octadepsipeptides and benzamides. However, resistance and efficacy problems are common for the few anthelmintic drug families currently available [2]. Furthermore, environmental impact and toxicity concerns have also restricted the use of some agricultural nematicides. These limitations prompt an urgent search for new and safe anthelmintics to be used in veterinary, agricultural, and medical applications. 

One of the main obstacles when developing new anthelmintic agents is the lack of efficient screening methods to assess the activity of compound libraries. The free-living nematode *C. elegans*, a widely used model in biology, is currently recognized as a cost-effective alternative for high-throughput screenings (HTS) of either synthetic or natural product libraries in nematicide discovery initiatives [3]. Despite not being a parasite, *C. elegans* is closely related to parasitic nematodes of the phylogenetic clade IV [4], which includes two of the most pathogenic nematodes of ruminants, such as *Haemonchus contortus* and *Cooperia* spp. [5].

Importantly, key studies have shown the utility of *C. elegans* as a surrogate parasitic nematode for drug discovery. As a proof of concept, independent screens of the compound library Pathogen Box using either *H. contortus* or *C. elegans* led to identification of the same compound as the nematicide [6,7]. A milestone study that screened 67,012 commercially available small drug-like molecules has shown that a compound active against *C. elegans* is at least 15 times more likely to be active against parasitic nematodes compared with a randomly chosen molecule [8]. This latter study identified *ortho*-substituted benzamides as a new family of compounds with excellent nematicidal activity. These compounds (depicted in Figure 1) are active against *C. elegans* and parasitic nematodes, with low toxicity in mammalian cells and animal models [8,9]. Reverse genetic experiments showed that these benzamides kill nematodes by inhibiting mitochondrial complex II [9], a key constituent of both the tricarboxylic acid cycle and the respiratory electron transport chain. X-ray structures obtained for *Ascaris suum* complex II co-crystallized with Wact-11-related compounds support this target as the main mechanism of nematicide action [10]. Furthermore, Fluopyram, a fungicide used in crops and recently repurposed as an agricultural nematicide, also belongs to this structural family [11]. As shown in Figure 1, the structural features of these compounds include the *o*-substituted benzamide ring (A), an aliphatic linker (B) and the *o*,*p*-substituted ring (C).

The structural diversity of the libraries previously screened includes benzamides with a vast number of substitution patterns at rings (A) and (C) and different linker chains. Recently, many other benzamide and aryl formamide derivatives have been described as nematicides, highlighting the importance of this scaffold [12]. Nevertheless, compounds with replacement of the amide group in active benzamides were not present in the libraries and, to the best of our knowledge, have not been reported in the literature. Bioisosterism, understood as the replacement of a group by another that shows a similar interaction potential with biomolecular targets, can drive hit-to-lead optimization by refining promising hits to create more potent and selective candidates [13]. Therefore, it emerged as a suitable design strategy to add a new dimension to the chemical space for this family of compounds. 

Various bioisosteric replacements of the amide group have been reported in the literature and web databases [14,15]. Each bioisostere preserves some of the geometrical and electronic features of the amide group while tuning others, allowing for the identification of features that are essential for the activity. The bioisosteres explored in this work were chosen according to the rate of success of each modification in other systems, along with information retrieved from structure-based in silico models (see Figure 2). 

Benzamides (such as Fluopyram, Fluopimomide) were originally used as fungicides and were recently described as nematicidal agents [12]. However, to the best of our knowledge, the activity of these compounds against platyhelminths has not yet been explored. The proposed molecular target of benzamides (complex II) uses the lipid rhodoquinone (RQ) as an electron transporter in both platyhelminths and nematodes. Furthermore, complex II has also been put forward as a promising target for *Echinococcus* spp. [16]. The presence of an RQ-dependent respiratory chain in both parasitic worm phyla opens the possibility for benzamides to be proposed as pan-anthelminthic compounds. 

This article describes the design, synthesis and evaluation of a set of 13 compounds derived from benzamides, **1a** (Wact-11) and **1b** (Wact-11p), described by Burns et al. as nematicides [8] and used in this study as references for activity assessment. In silico studies were first carried out to allow a structure-based bioisosterism design. Thirteen new benzamide bioisosteres were then prepared and evaluated in *C. elegans* in order to explore the effect of the amide group replacement in the nematicide activity, gaining insights into their SARs. Compounds showing the best results against *C. elegans* were also evaluated against *E. granulosus* in a protoscolex survival assay, aiming to broaden the anthelminthic activity to a new phylum.

## 2. Results and Discussion

### 2.1. Chemistry: Rational and Synthesis

#### 2.1.1. In Silico Design

In order to select which bioisosteric replacements would be prioritized for synthesis and biological evaluation, we took advantage of several computational tools. Apart from literature revision, we resorted to the Swissbioisostere database [17], a user-friendly and comprehensive search tool for bioisosteric replacements and their reported impact on different drug targets. 

A structural analysis of nematode complex II was explored as another input for bioisosteric design, using an approach based on docking experiments. Structure-based in silico methodologies were recently applied to discover novel benzamide-related nematicides [18]. There are several crystallographic structures of *Ascaris suum* hypoxic complex II deposited in the PDB database, co-crystallized with their natural substrate (rhodoquinone) and inhibitors, most of them structurally related to benzamides [19]. These experimentally determined complex II structures were used as templates to build homology models of *C. elegans* complex II and to test the capability of the docking protocol to replicate the experimental pose of the ligands. After optimization of the docking protocol (see Appendix A), previously reported inhibitor libraries were tested on *A. suum* and *C. elegans* complex II [8,19], as well as our proposed analogs. The docking methodology proved useful to check for notorious steric hindrances, possible unoccupied subpockets and relevant ligand-receptor interactions as detailed below. However, docking scores obtained from different scoring functions showed little correlation with the reported experimental IC_50_ values, thus limiting the utility of docking as a quantitative predictor of complex II inhibition or nematicidal potential and suggesting the need for more precise estimators of the interaction energy. 

#### 2.1.2. Synthesis

Benzamides **1a** (Wact-11) and **1b** (Wact-11p), reported by Burns et al. as nematicides [8], were first prepared as positive controls for activity assessment. The synthesis of benzamides **1a** and **1b** was carried out starting from 2-trifluoromethylbenzoic acid or 2-iodobenzoic acid using oxalyl chloride to generate the corresponding acid chloride in situ and was then treated with the corresponding amine to yield **1a** (R = 58%) and **1b** (R = 76%) respectively. 

Ester, thioamide and selenoamide derivatives. 

Classic amide bioisosteres were first prepared. Despite its metabolic lability, the ester group can help to reveal the role played by the amide nitrogen in the nematicidal activity of benzamides. Analysis of crystallographic structures of *A. suum* complex II and homology models built for *C. elegans* complex II with benzamides co-crystallized or docked did not show any evident interaction between the NH and complex II amino acid residues [19], thus raising the question if this amide feature is essential for activity. Ester derivative **2** was prepared by treating 2-iodobenzoic acid with DCC in the presence of DMAP and the corresponding alcohol 2-(4-chlorophenyl)ethanol); see Figure 1I.

Sulfur is a classical isostere of oxygen since both atoms belong to the same group. The impact of this isosteric substitution is generally an increase in lipophilicity with only subtle geometric modifications [15,20]. In the context of amides, this replacement has implications for stability and potential interactions with biomolecular receptors. The hydrogen bond acceptor capacity of thiocarbonyl is lower than that of carbonyl, while the thioamide NH bond is more acidic and prone to donor hydrogen bonds [21]. These features make thioamides an appropriate replacement for SAR studies, particularly to analyze the contribution of hydrogen bond donor and acceptor patterns to the activity of the amide analog compounds. Thioamide **3** was prepared from amide **1b** by thionation using Lawesson’s reagent under reflux [22]; see Figure 1II.

Selenium is receiving growing attention in medicinal chemistry, as an oxygen and sulfur isostere. The selenoamide group has been incorporated into antibacterial [23], antioxidant [24] and peptidomimetic compounds [25]. Despite being less prone to accept hydrogen bonds than sulfur [26], selenium can engage in chalcogen bonds, non-covalent interactions whose importance have not yet been fully exploited in drug design [27]. Selenoamide derivative **4** was prepared starting from amide **1b** using Woollins’ reagent as the selenating agent, according to the procedure originally published [28]; see Figure 1II.

*N*-alkyl amide and *N*-alkyl thioamide. 

Alkylation as a drug design strategy is mainly used to increase lipophilicity, incorporate hydrophobic interaction sites or decrease the rate of metabolism [29]. The rationale for exploring the substitution of the hydrogen atom by alkyl chains in the Wact-11 family arises from the visual inspection of homology models built for *C. elegans* complex II and docking experiments for benzamide derivatives (see in silico studies section for further details). In those models, a cavity defined by residues Trp67 (Chain C), Pro211 (Chain B) and Ile260 (Chain B) remains unoccupied when benzamides such as Wact-11 are docked in the quinone pocket of complex II. The identities of the residues of this sub-pocket are conserved in the X-ray structures of *A. suum* complex II co-crystallized with benzamide compounds, suggesting that the cavity could also be available in other parasitic nematodes. The dimensions of this sub-pocket made us wonder if methyl or alkyl N-substituted amides could fill the cavity. When *N*-methyl and *N*-ethyl derivatives of benzamides were docked in the benzoquinone pocket, they accommodated favorably, without clashes and preserving the pose features of their secondary amide analogs. 

In order to explore this substitution, *N*-alkylamide derivatives (**5a**–**c**) and *N*-methylthioamide **6** were prepared in good yields by treating amides and thioamide with NaH and then adding the corresponding alkyl iodide; see Table 1. 

Urea, sulfonamide and triazole derivatives.

Urea is one of the most-used amide bond replacements in medicinal chemistry programs. Similarities between the urea and amide groups are easily recognizable: the amide H-bond donor and acceptor are conserved, and potential additional H-bond interactions are incorporated with the extra NH. Ureas are usually employed to lengthen the distance between moieties while maintaining the planar geometry [30,31]. 

Urea derivative **7** was prepared by treating 2-(4-chlorophenyl)ethan-1-amine with 1-iodo-2-isocyanato-benzene, generated by the reaction of 2-iodoaniline with triphosgene; see Table 2. Sulfonamide, another amide bioisostere, increases hydrophobicity and solubility while keeping the hydrogen bond interaction pattern of the parent compound [14]. This is of special interest when designing protease inhibitors, as sulfonyl tetrahedral geometry mimics that of the hydrolysis transition state [32]. Derivatives **8a**–**c** were readily prepared by treating 2-(trifluoromethyl)-benzene-sulfonyl chloride with the corresponding amines, which gave excellent yields of the sulfonamide analogs; see Table 2.

The last group selected for amide replacement was the triazole group, recognized as a non-classical bioisostere of amides, particularly in the context of peptidomimetics [33]. Recent applications of this bioisosteric replacement include HIV protease inhibitors [34], cyclic tyrosinase inhibitors [35], transmembrane conductance modulators for cystic fibrosis [36], and BACE1 inhibitors [37], among others. The triazole group preserves the planar geometry and a similar distribution of the hydrogen bond acceptor and donor features of the amide bond. Amide and triazole mainly differ in their lipophilicity (significantly higher for the triazole ring) and their length (triazole being ~1 Å longer).

Triazole derivatives were designed as amide analogs, including a triazole bioisostere of flutolanil (**9c**), a benzamide compound that inhibits *A. suum* complex II [19]. Triazoles **9a**–**c** were prepared via Huisgen copper-catalyzed cycloaddition [38] between alkynes **10** and azides **11**, as shown in Table 3. Alkyne **10a** is commercially available, while the *ortho*-substituted alkyne **10b** was prepared from 1-iodo-2-(trifluoromethyl) benzene using a Sonogashira reaction [39]. Aliphatic azide **11a** was prepared using the Goddard–Borger reagent [40], while aromatic azide **11b** was prepared by diazotization of the corresponding aniline under conventional conditions [41]. Due to stability issues during **10b** trimethylsilyl (TMS) deprotection, a one-pot strategy for TMS cleavage (with K_2_CO_3_) and the Huisgen cycloaddition was carried out, taking advantage of the robustness of the latter.

### 2.2. Biological Evaluation

The compound library was screened using the Phylumtech WMicrotracker ONE monitoring device to study nematicidal activity against *C. elegans*. This equipment measures motility in a simple and high-throughput infrared-based automated assay [7,42]. We used a recently adapted assay described by our group with increased sensitivity for benzamides using L1 instead of L4 [43].

Compounds **1a** (Wact-11) and **1b** (Wact-11p) were first assayed in order to set a reference value to compare with the prepared bioisosteres.

#### 2.2.1. *C. elegans* L1 Assay

Using the L1 larvae-adapted assay, **1a** (Wact-11) and **1b** (Wact-11p) were assayed against *C. elegans*. For both compounds, a motility reduction of 100% was observed at 10 μM after 20 h. Thus, these conditions (10 µM concentration and 20 h) were used to perform the initial screening of the synthesized compounds. 

The results for Wact-11 bioisosteres **2**–**4**, **5a**–**c**, **6**, **7** and **8b** are shown in Table 4, and they allowed us to directly compare the effect of bioisosteric substitution on nematicidal activity. Compounds **8a**, **8c** and **9a**–**c** were also evaluated against *C. elegans* but showed no activity, Appendix A.

As shown in Table 4, bioisosteric replacement using *N*-alkylamides **5a**–**c** and sulphonamide **8** showed no significant activity against *C. elegans* (entries 6–8 and 11). *N*-methylthioamide **6** and urea **7** gave about 50% motility reduction (59% and 47%, respectively, for entries 9 and 10). The best results were obtained for thioamide **3** (92%) and selenoamide **4** (100%), presenting a motility reduction similar to the reference compounds **1a** and **1b** (entries 4 and 5). 

The bioisosteric replacements of oxygen by sulfur or selenium were the only variations that retained the activity against *C. elegans*. This could be explained since these modifications preserve the amide geometry, highlighting the importance of the amide group in these compounds. EC_50_ for both compounds were determined, and the results are shown in Figure 3.

The EC_50_ values of **3** and **4** are in the same sub-10 μM range as the reference **1b** but are less active, and thioamide **3** is 2.6 times less active than **1b**. This decrease could be due to the lower H-bond acceptor capacity of S compared to O. Since the thioamide NH H-bond donor capacity is stronger than the amide, the lower activity of **3** suggests that this property does not play a crucial role in the activity. Conversely, the best result was obtained for selenoamide **4** with EC_50_ = 2.4 ± 0.5 μM, compared to the reference compound **1b** (EC_50_ = 1.7 ± 0.4 μM). This result is interesting and unexpected, since it breaks the tendency observed for thioamide **3** compared to amide **1b**. As selenium has an even lower capacity than sulfur to establish H-bonds, the foreseen activity should have been lower. The result observed suggests another kind of interaction playing a role in nematicide activity for this compound or the possibility of a different target.

#### 2.2.2. *Echinococcus granulosus* Protoscolex Assay

As complex II is also present in plathelminths and, similarly to nematodes, acts as fumarate reductase giving electrons to rhodoquinone, benzamide **1a** and the most active bioisosteres **3** and **4** were selected to be tested against the tapeworm *E. granulosus*, the causative agent of cystic echinococcosis (hydatid disease).

In particular, we assessed viability of protoscolex, the larval worm present within the hydatid cyst in the intermediate host (cattle and accidentally humans), and the infective stage for canids, the definitive host that ingests cyst-containing viscera from cattle. Protoscolex is also responsible for secondary hydatid disease in the intermediate hosts, when a cyst within a viscera is accidentally ruptured. Benzamide **1a** (Wact-11), thioamide **3** and selenoamide **4** were tested in a protoscolex survival assay using a vital dye. The results are shown in Figure 4.

Compound **1a** (Wact-11) exhibited an EC_50_ value of 10 μM. In contrast, thioamide **3** and selenoamide **4** did not exhibit significant differences compared to negative control at 20 μM. These results indicate that benzamides are active not only against nematodes, but also against platyhelminths, suggesting a pan-helminthic potential for compounds belonging to this family.

### 2.3. In Silico Studies

Both ligand-based and structure-based computer-aided drug discovery approaches were performed. The former was largely based on the compound database screened by Burns et al. from which SAR, QSAR and pharmacophore models were built using MOE-integrated capabilities. In general, those models showed poor discriminative capacity when applied to compound databases including *C. elegans* complex II inhibitors and inactive molecules during the validation stage. This was probably due to the fact that all complex II nematicides found by Burns et al. are benzamides, biasing the models to this structural feature. This strongly limited the applicability of these models to the derivatives designed.

The structure-based approach took advantage of the crystallographic structures of *A. suum* complex II available on PDB, several of which are co-crystallized with benzamides. *C. elegans* complex II was modeled by homology using these structures as templates. Docking studies of the reference inhibitors of complex II and the designed derivatives were performed using the software MOE. During the cross-validation stage, the docking protocol showed high reliability when predicting the crystallographic pose of benzamides in *A. suum* complex II structures. However, the binding affinity estimated through a set of scoring functions showed a low correlation with the EC_50_ values reported for the inhibitors, thus limiting the utility of docking in the prediction of complex II inhibition or nematicide potential of new compounds. Moreover, the absence of a membrane environment in the simulation could be another limitation in the prediction of the model. In this sense, an additional modeling considering a more physiological environment was performed (see Section 2.3.2). 

Despite these limitations, the analysis of the proposed binding mode of benzamide nematicides proved to be useful for the design of derivatives and could be a source of SAR hypothesis and of possible explanations for the loss of potency of derivatives examined. 

#### 2.3.1. Docking Analysis: Benzamides Interaction with Nematode Complex II

Consistently with previous reports [8,9,19], our docking analysis showed that benzamides interact with nematode’s complex II mainly through the amidic carbonyl and ring A moieties. The amidic carbonyl can establish H-bonds with the near residues Trp215 of subunit B and Tyr96 of subunit D (numbering corresponding to *C. elegans*, Trp197 and Tyr107 D in *A. suum* complex II). Ring A establishes a key interaction with Arg74 of subunit C (Arg76 C in *A. suum*), consisting of an H-bond between the guanidino NH and the halogen of the *ortho* group of the benzamide ring, along with an ion–π interaction with the ring (see interaction diagram in Figure 5 for **1a** and Appendix A for compounds **3** and **4**). 

We also observed that the benzamide ring interaction is strengthened by an electrostatic gradient established by the electron-withdrawing group in *ortho* position (-CF_3_). Both the docking poses predicted and the experimental crystallographic structures showed that the amide carbonyl and the *ortho* substituting group are parallel and equiplanar, reinforcing the electronic effect over the ring. The requirement for this group in this position can be explained by the high complementarity between ligand and receptor electrostatic surfaces at this level, as shown in Figure 6a,b.

The receptor surface contacting the *o*-substituent and carbonyl side of the ring is rich in nitrogen-containing residues and amino groups from the polypeptide chain backbone, while the surface of the opposite side predominantly exposes oxygen atoms to the pocket, inducing an electronegative environment and thus complementing the uneven electronic distribution of the ligand benzamide ring (Figure 6b). All these structural features are predicted to be present in all nematodes, as phylogenetic analysis showed that the crucial residues are highly conserved along the phylum. 

The predicted binding mode of all bioisostere derivatives containing an *o*-substituted ring A was analogous to that of benzamides (see Appendix A). Ring A seems to be a key determinant for the binding mode, mainly due to halogen and hydrophobic interactions with the *o*-substituent group (CF_3_ or I) and the strong shape complementarity in this region of the pocket (shown in Appendix A).

The analysis of the *E. granulosus* complex II subunits sequence showed that most of the benzoquinone binding sites (including all key residues involved in benzamide interaction with the target) are conserved, despite the low global identity in the two cytochrome-binding subunits (C and D) of the complex. Furthermore, structure analysis over a homology model of *E. granulosus* complex II and docking studies suggest that the pocket geometry is similar to that of nematode complex II (data not shown).

#### 2.3.2. Simulated Mitochondrial Membrane

While docking experiments were conducted in a simulated vacuum phase, we became interested in the possible effects of the physiological environment in the ligand-receptor interaction for benzamide derivatives, particularly for conducting molecular dynamics experiments on the system. Nematode complex II structures, both from PDB and our homology modeling, were embedded in a simulated mitochondrial inner membrane, built using CHARM-GUI [44] membrane builder module and OPM membrane location predictor [45]. The experiments were conducted for compound **1a** (Figure 6c), Fluopyram (see Appendix A) and analogs. In these models, the quinone pocket (where benzamide derivatives bind to the complex) is fully embedded in the membrane. An interesting observation is the proximity of the pocket with the polar heads of the lipids forming the layer facing to the intermembrane space. This environment has a relatively high potential for establishing interactions through the polar phosphates and amine groups present. This could, in part, explain a SAR observation regarding the C ring of benzamides, in which halogen or CF_3_ di-substitution seems to increase the potency of the inhibitors with respect to the mono-substituted analogs. For example, the compound Wact-11f is identical to Wact-11 with the only addition of a 2-Cl moiety on the C ring, which according to Burns et al., has an LD_100_ against *C. elegans* of 0.469 μM, while the LD_100_ of monosubstituted Wact-11 is of 7.5 μM [8].

The structural observations herein presented are in strong agreement with the observed SAR for benzamide derivatives, either for those prepared in this work or previously screened by other groups. These results shed light on the causes of the privileged status of the benzamide moiety in helminth complex II inhibition.

## 3. Materials and Methods

### 3.1. General Experimental Parameters

All reactions and chromatographic separations were monitored by analytical thin-layer chromatography (TLC) 0.25 mm Silica gel plastic sheets (Macherey–Nagel, Polygram^®^ SIL G/UV 254). TLC plates were analyzed under 254 nm UV light or using iodine vapor. Flash chromatography on Silica gel 60 (J. T. Baker, 40 mm average particle diameter) was used to purify the crude reaction mixtures. Microwave-assisted synthesis was performed in a CEM Discover reactor. NMR spectra were recorded at 400 MHz (^1^H NMR) and 100 MHz (^13^C NMR) using a Brucker Avance NEO-400 spectrometer at 21 °C. Chemical shifts (δ) are reported as follows: chemical shift (multiplicity (s: singlet, d: doublet, t: triplet, q: quartet, quint: quintet, m: multiplet), coupling constant, integration) using TMS as reference. When two conformers are observed, (*) denotes the signal corresponding to the minor one. High-resolution mass spectrometry experiments (HRMS) were measured on a MicroTOF-Q-Bruker Daltronics using electrospray ionization. Melting points were determined using a Fisher-Jones Melting Point apparatus. All solvents were purified according to literature procedures. All yields refer to isolated compounds after the final purification process, unless otherwise stated.

### 3.2. Compound Synthesis

Wact compounds **1a** and **1b** were prepared as previously described. ^1^H- and ^13^C-NMR data are according to those reported in the literature [46]. Compounds **9a**, **10b**, **11a**, **11b** and **11c** are described in the literature and the spectroscopic data are according to those reported (shown in Appendix A).

4-chlorophenethyl 2-iodobenzoate (**2**). To a solution of 2-iodobenzoic acid (200 mg, 0.8 mmol) in dry CH_2_Cl_2_ (10 mL) under N_2_ was added 2-(4-chlorophenyl)ethan-1-ol (131 mg, 0.84 mmol), DCC (185 mg, 0.9 mmol) and DMAP (10 mg, 0.08 mmol). The mixture was stirred at room temperature for 2 h, then the solvent was removed under reduced pressure and the residue was purified by flash column chromatography on silica gel (EtOAc/*n*-Hex (1:6)) to yield compound **2** (230 mg, 59%) as a white solid: MP 56 °C. ^1^H NMR (CDCl_3_): δ 3.06 (t, *J* = 6.9 Hz, 2H), 4.52 (t, *J* = 6.9 Hz, 1H), 7.26 (m, 6H), 7.69 (dd, *J* = 7.8, 1.7 Hz, 1H), 7.97 (d, *J* = 8.0 Hz, 1H). ^13^C NMR (CDCl_3_): δ 34.4, 65.8, 94.2, 128.0, 128.7, 130.4, 130.9, 132.5, 132.7, 135.0, 132.7, 136.2, 141.4, 166.4; HRMS (ESI) calcd. for C_15_H_12_ClIO_2_ 408.9463 [M + Na]^+^, found 408.9469. 

*N*-(4-chlorophenethyl)-2-iodobenzothioamide (**3**). To a solution of **1b** (Wact11p) (100 mg, 0.25 mol) in THF (4 mL) under N_2_ atmosphere, was added Lawesson’s reagent (121 mg, 0.3 mmol). The mixture was stirred at room temperature for 24 h and the solvent was removed under reduced pressure. The residue was poured into a saturated solution of NaHCO_3_ (20 mL) and extracted with EtOAc (3 × 20 mL). The reaction crude was purified by flash column chromatography on silica gel (EtOAc/*n*-Hex (1:5)) to yield **3** (61 mg, 61%) as a white solid: MP = 143 °C. ^1^H NMR (CDCl_3_): δ 3.08 (t, *J* = 7.2 Hz, 2H), 4.06 (td, *J* = 7.2, 5.7 Hz, 2H), 7.02 (ddd, *J* = 7.9, 6.5, 2.6 Hz, 1H), 7.30 (m, 7H), 7.79 (d, *J* = 7.7 Hz, 1H). ^13^C NMR (CDCl_3_): δ 33.0, 46.8, 92.0, 128.2, 128.4, 129.0, 130.2, 130.3, 132.8, 136.5, 139.7, 148.2, 201.5. HRMS (ESI) calcd. for C_15_H_13_ClINS 423.9390 [M + Na]^+^, found 423.9392.

*N*-(4-chlorophenethyl)-2-iodobenzoselenoamide (**4**). To a solution of **1b** (Wact11p) (100 mg, 0.25 mmol) in dry toluene (2 mL) under N_2_ atmosphere was added Woolins’ reagent (46 mg, 0.09 mmol). The mixture was refluxed for 4 h, and the solvent was removed under reduced pressure. The residue was purified by flash column chromatography on silica gel (EtOAc/*n*-Hex (1:5)) to yield **4** (88 mg, 70%) as a yellow solid: MP = 209 °C; Rf (EtOAc/*n*-Hex (1:2)) = 0.7. ^1^H NMR (CDCl_3_): δ 3.14 (t, *J* = 7.3 Hz, 2H), 4.07 (m, 2H), 7.00 (td, *J* = 7.6, 1.8 Hz, 1H), 7.29 (m, 5H), 7.40 (dd, *J* = 7.7, 1.8 Hz, 1H), 7.77 (d, *J* = 8.0 Hz, 1H), 7.89 (s, 1H). ^13^C NMR (CDCl_3_): δ 32.7, 50.2, 90.8, 127.7, 128.2, 130.2, 130.3, 132.9, 136.2, 139.7, 150.8, 207.3. HRMS (ESI) calcd. for C_15_H_12_ClINSe 447.8868 [M − H]^−^, found 447.8867.

*N*-(4-chlorophenethyl)-*N*-methyl-2-(trifluoromethyl)benzamide (**5a**). To a suspension of NaH (23.5 mg, 0.35 mmol) in DMF (1 mL) under N_2_ atmosphere, a solution of **1a** (Wact11) (100 mg, 0.31 mmol) in DMF (1 mL) was added dropwise. When gas evolution ceased, methyl iodide (54 mg, 0.38 mmol) was added, and the mixture was stirred at room temperature for 4 h. The resulting mixture was poured into water, neutralized with HCl 5% and extracted with EtOAc (3 × 20 mL). The combined organic layers were dried and filtered. The solvent was removed under reduced pressure and the crude was purified by flash column chromatography on silica gel (EtOAc/*n*-Hex (1:4)) to give **5a** as a mixture of *Z*/*E* isomers (yellow oil, 42 mg, 40%). ^1^H NMR (CDCl_3_): δ 2.69/3.15* (2x s, 3H), 2.75* (td, *J* = 7.1, 6.7, 3.3 Hz, 2H)/2.94 (m, 2H), 3.25* (t, *J* = 7.3 Hz, 2H)/3.75 (m, 2H), 6.82 (m, 1H), 7.24 (m, 4H), 7.47 (m, 1H), 7.58 (t, 1H), 7.67 (m, 1H). ^13^C NMR (CDCl_3_): δ 32.5/37.4*, 32.7/33.6*, 37.4, 48.8/52.5*, 123.68 (q, *J_CF_* = 273.9 Hz), 126.4/126.6 (2x q, *J_CF_* = 4.5 Hz), 127.2/127.4*, 128.7/128.8*, 129.0, 130.1*/130.2, 131.2, 132.0, 132.2/132.7*, 132.3, 134.8*/135.4 (2x q, *J_CF_* = 2.4 Hz), 136.3*/137.3, 168.7/168.9*. HRMS (ESI) calcd. for C_17_H_15_ClF_3_NO 364.0692 [M + Na]^+^; found 364.0699.

*N*-(4-chlorophenethyl)-*N*-ethyl-2-iodobenzamide (**5b**). Compound was obtained following the procedure described for **5a** starting from **1b** (Wact11p) (100 mg, 0.25 mmol) and ethyl bromide (34 mg, 0.31 mmol) to yield **5b** as a mixture of *Z*/*E* isomers (yellow oil, 82 mg, 76%). Rf (EtOAc/*n*-Hex (1:1)) = 0.85. ^1^H NMR (CDCl_3_): δ 1.05/1.33 (2x t, *J* = 7.1 Hz), 2.73 (dt, *J* = 21.4, 7.9 Hz, 2H)/3.03 (m, 2H), 3.10* (m, 2H)/ 3.69 (bd, *J* = 163.1 Hz, 2H), 3.23 (2x m, 1H), 3.34 (m, 1H)/4.00 (m, 1H), 6.82*/6.83 (2x t, *J* = 7.7 Hz, 2H), 7.06 (2x td, *J* = 9.6, 7.8 Hz, 1H), 7.17 (2x m, 2H), 7.28 (m, 4H), 7.38 (2x t, *J* = 7.5 Hz, 1H). ^13^C NMR (CDCl_3_): δ 12.3*/13.9, 33.0/34.4*, 39.6*/43.8, 46.1/49.6*, 92.7, 126.9/127.2*, 128.2*/128.3, 128.7/128.8*, 130.0*/130.1, 130.1*/130.3, 132.2/132.5*, 136.4*/137.6, 139.0*/139.2, 142.3*/142.6, 170.4*/170.5. HRMS (ESI) calcd. for C_17_H_17_ClINO 435.9941 [M + Na]^+^; found 435.9937. 

*N*-(4-chlorophenethyl)-2-iodo-*N*-isobutylbenzamide (**5c**). Obtained following the procedure described for **5a** starting from **1b** (Wact11p) (100 mg, 0.25 mmol) and 1-bromo-2-methylpropane (42 mg, 0.31 mmol) to yield **5c** as a mixture of *Z*/*E* isomers (yellow oil, 70 mg, 63%). Rf (EtOAc/*n*-Hex (1:1)) = 0.85. ^1^H NMR (CDCl_3_): δ 0.78 (dd, *J* = 17.8, 6.6 Hz, 3H)/ 1.06* (d, *J* = 6.7 Hz, 3H), 1.87/2.23 (2 x dquint, *J* = 13.5, 6.8 Hz, 1H), 2.68* (m, 1H)/ 2.98 (dt, *J* = 13.4, 6.8 Hz, 1H), 2.79*/3.12 (m, 1H), 2.83/3.26 (2x m, 1H) 3.06 /3.11* (2x m, 1H), 3.06/3.68* (2x m, 1H), 3.26*/3.42 (2x m, 1H), 3.33*/3.97 (2x m, 1H), 6.78*/6.80 (2x s, 1H), 6.95 (m, 1H), 7.06 (m, 1H), 7.17 (m, 1H), 7.27 (m, 2H), 7.36 (m, 1H), 7.82 (ddd, *J* = 7.7, 6.0, 1.1 Hz, 1H). ^13^C NMR (CDCl_3_): δ 19.8*/20.8, 20.4*/20.9, 46.7/50.9*, 52.1*/56.1, 92.6*/93.0, 127.5, 128.1, 128.08/128.13*, 128.7/128.8*, 130.97/130.02*, 130.1, 130.2, 132.2, 132.5, 136.4, 137.6, 139.1/139.3*, 142.3/142.8*, 170.9*/171.0. HRMS (ESI) calcd. for C_19_H_21_ClINO 464.0254 [M]^+^; found 464.0254. 

*N*-(4-chlorophenethyl)-2-iodo-*N*-methylbenzothioamide (**6**). The title compound was obtained following the same procedure for **3** starting from **1b** (Wact11p) (100 mg, 0.29 mmol) and Lawesson’s reagent (142 mg, 0.35 mmol), yielding compound **6** as a mixture of *Z*/*E* isomers (yellow oil, 56 mg, 47%). Rf (EtOAc/*n*-Hex (1:4)) = 0.9. ^1^H NMR (CDCl_3_): δ 2.84* (m, 2H)/3.07 (m, 1H) + 3.32 (ddd, *J* = 13.3, 10.2, 5.9 Hz, 1H), 3.03 (s, 3H)/3.58* (s, 3H), 3.54* (m, 2H)/ 4.06 (ddd, *J* = 12.8, 10.2, 5.6 Hz, 1H) + 4.45 (ddd, *J* = 12.6, 10.2, 5.9 Hz, 1H), 6.80* (dd, *J* = 7.7, 1.6 Hz, 1H)/ 6.84 (d, *J* = 8.4 Hz, 1H), 6.98* (m, 1H), 7.12 (m, 2H), 7.3 (s, 3H), 7.37*/7.38 (2x t, *J* = 7.5 Hz, 1H), 7.78 (m, 1H). ^13^C NMR (CDCl_3_): δ 30.8/33.9*, 40.4*/41.8, 56.7/57.0*, 92.2/92.7*, 126.2/127.0*, 128.3*/128.7, 128.8, 128.9, 129.15/129.18, 130.1*/130.3, 132.5/132.9*, 135.6*/136.7, 139.1*/139.4, 147.4*/147.8, 200.2*/200.7. HRMS (ESI) calcd. for C_16_H_15_ClINS 437.9556 [M + Na]^+^, found 437.9556. 

1-(4-chlorophenethyl)-3-(2-iodophenyl)urea. (**7**) To a solution of triphosgene (54 mg, 0.2 mmol) in dry CH_2_Cl_2_ (2.5 mL) under N_2_ atmosphere at 0 °C, 2-iodoaniline (43 mg, 0.2 mmol) and Et_3_N (47 mg, 0.92) were added. The mixture was then stirred at room temperature and monitored by TLC until 2-iodoaniline was consumed. The mixture was then cooled to 0 °C, and a solution of 2-(4-chlorophenyl)ethan-1-amine (0.89 mg, 0.56 mmol) and Et_3_N (47 mg, 0.92 mmol) in dry CH_2_Cl_2_ (2.5 mL) was added. The reaction was stirred at room temperature overnight. Then, the mixture was poured into water, neutralized with HCl 5% and extracted with EtOAc (3 × 20 mL). The crude was purified by flash column chromatography on silica gel (EtOAc/*n*-Hex (1:6)) to yield **7** (20 mg, 11%) as a white solid: MP = 172 °C; Rf (CH_2_Cl_2_) = 0.3. ^1^H-NMR (CO(CD_3_)_2_) δ 2.86 (t, *J* = 7.1 Hz, 2H), 3.48 (td, *J* = 7.0, 5.8 Hz, 2H), 6.58 (bs, 1H), 7.17 (bs, 1H), 7.32 (m, 5H), 7.78 (dd, *J* = 7.9, 1.5 Hz, 1H), 8.06 (dd, *J* = 8.3, 1.6 Hz, 1H). ^13^C NMR (CO(CD_3_)_2_) δ 37.2, 42.9, 90.7, 123.8, 125.9, 130.1, 130.4, 132.4, 133.2, 140.5, 140.7, 142.6, 156.5. HRMS (ESI) calcd. for C_15_H_14_ClIN_2_ONa 422.9737 [M + Na]^+^; found 422.9751.

*N*-(3-isopropoxyphenyl)-2-(trifluoromethyl)benzenesulphonamide (**8a**) To a solution of 2-(trifluoromethyl)benzenesulfonyl chloride (100 mg, 0.41 mmol) in dry CH_2_Cl_2_ (1 mL) under N_2_ atmosphere at 0 °C, Et_3_N (73 mg, 0.71 mmol) and 3-isopropoxyaniline (62 mg, 0.41 mmol) were added. The mixture was stirred at room temperature overnight, then poured into water, acidified to pH 5 and extracted with EtOAc (3 × 20 mL). The combined organic layers were dried with Na_2_SO_4_, filtered and removed under reduced pressure. The crude was purified by flash column chromatography on silica gel (CH_2_Cl_2_/*n*-Hex (1:2)) to yield **8a** (124 mg, 84%), as a dark solid: MP =108 °C; Rf (EtOAc/*n*-Hex (1:4)) = 0.6. ^1^H NMR (CDCl_3_): δ 1.26 (s, 3H), 1.27 (s, 3H), 4.45 (hept, *J* = 6.0 Hz, 1H), 6.57 (ddd, *J* = 7.9, 2.1, 0.9 Hz, 1H), 6.62 (ddd, *J* = 8.3, 2.4, 0.9 Hz, 1H), 6.62 (ddd, *J* = 8.3, 2.4, 0.9 Hz, 1H), 6.66 (t, *J* = 2.3 Hz, 1H) 6.74 (bd, *J* = 8.2 Hz, 1H)(NH), 7.07 (t, *J* = 8.1 Hz, 1H), 7.57 (td, *J* = 7.8, 1.4 Hz, 1H), 7.64 (t, *J* = 7.6 Hz, 1H), 7.86 (d, *J* = 7.7 Hz, 1H). ^13^C NMR (CDCl_3_): δ 21.7, 70.3, 109.5, 123.0 (q, *J_CF_* = 273.9 Hz), 127.7 (q, *J_CF_* = 32.9 Hz), 128.5 (q, *J_CF_* = 6.4 Hz), 130.1, 132.2, 132.5, 133.1, 136.8, 137.2, 158.7. HRMS (ESI) calcd. for C_16_H_11_ClF_3_NO_2_S 382.0702 [M]^+^; found 382.0703. 

*N*-(4-chlorobenzyl)-2-(trifluoromethyl)benzensulfonamide (**8b**). Compound **8b** was obtained following the same procedure for **8a** starting from 2-(trifluoromethyl) benzenesulfonyl chloride (100 mg, 0.41 mmol), and *p*-chlorobenzylamine (58 mg, 0.41 mmol) to yield **8b** (152 mg, 99%) as a white solid: MP = 84 °C; Rf (EtOAc/*n*-Hex (1:4)) = 0.45. ^1^H NMR (CDCl_3_): δ 4.15 (d, *J* = 6.2 Hz, 2H), 5.09 (t, *J* = 6.4 Hz, NH), 7.12 (m, 2H), 7.21 (m, 2H), 7.68 (m, 2H), 7.86 (m, 1H), 8.15 (m, 1H). ^13^C NMR (CDCl_3_): δ 46.7, 123.0 (q, *J_CF_
*= 273.8 Hz), 127.4 (q, *J_CF_* = 32.9 Hz), 128.5 (q, *J_CF_* = 6.3 Hz), 128.8, 131.7, 132.4, 132.5, 132.8, 133.7, 134.5, 138.6, 138.7. HRMS (ESI), calcd. for C_14_H_11_ClF_3_NO_2_SNa 372.0049 [M + Na]^+^; found 372.0059.

*N*-(4-chlorophenethyl)-2-(trifluoromethyl)benzenesulphonamide (**8c**). Compound **8c** was obtained following the same procedure for **8a** starting from 2-(trifluoromethyl) benzenesulfonyl chloride (100 mg, 0.41 mmol), Et_3_N (73 mg, 0.71 mmol) and 2-(4-chlorophenyletan-1-amine) (63.5 mg, 0.41 mmol) in CH_2_Cl_2_ (1 mL). The mixture was stirred at room temperature overnight, then poured into water, acidified to pH 5 and extracted with EtOAc (3 × 20 mL). Combined organic layers were dried with Na_2_SO_4_, filtered and removed under reduced pressure. The crude was purified by flash column chromatography on silica gel (CH_2_Cl_2_/*n*-Hex (1:2)) to yield **8c** (150 mg, 99%) as a white solid: MP = 52 °C; Rf (EtOAc/*n*-Hex (1:4)) = 0.4. ^1^H NMR (CDCl_3_): δ 2.75 (t, *J* = 7.0 Hz, 2H), 3.25 (q, *J* = 6.7 Hz, 2H), 4.75 (bt, *J* = 6.2 Hz, 1H)(NH), 6.99 (m, 2H), 7.18 (m, 2H), 7.69 (m, 2H), 7.84 (m, 1H), 8.15 (m, 1H). ^13^C NMR (CDCl_3_): δ 36.3, 44.4, 122.9 (q, *J_CF_* = 273.9 Hz), 127.3 (q, *J_CF_* = 32.9 Hz), 128.5 (q, *J_CF_* = 6.1 Hz), 128.8, 130.0, 131.4, 132.4, 132.6, 132.7, 135.9, 138.5. HRMS (ESI) calcd. for C_15_H_13_ClF_3_NO_2_SNa 386.0205 [M + Na]^+^; found 386.0215.

1-(4-chlorophenethyl)-4-(2-(trifluoromethyl)phenyl)-1H-1,2,3-triazole (**9b**). Compound **10a** (200 mg, 1.1 mmol) was dissolved in 4 mL of a *t*-BuOH/H_2_O (1:1) mixture. K_2_CO_3_ (110 mg, 0.72 mmol) was added and the mixture was stirred for 4 h. The mixture was neutralized with HCl 5% and then CuSO_4_·5H_2_O (4.2 mg, 0.02 mmol), sodium ascorbate (40 mg, 0.2 mmol) and **11b** (194.9 mg, 1.1 mmol) were added. The reaction was stirred overnight at rt, then poured into water (10 mL) and extracted with CH_2_Cl_2_ (3 × 20 mL). The combined organic layers were dried with Na_2_SO_4_, then filtered and the solvent was removed under reduced pressure. The crude was purified by flash column chromatography on silica gel (EtOAc/*n*-Hex (1:1)) to yield **9b** as a yellow oil (172 mg, 56%); Rf (EtOAc/*n*-Hex (1:1)) = 0.5. ^1^H NMR (CDCl_3_): δ 1.36 (s, 3 H), 1.38 (s, 3H), 4.63 (quint, *J* = 6.1 Hz, 1H), 6.94 (ddd, *J* = 8.2, 2.5, 1.0 Hz, 1H), 7.27 (m, 1H), 7.36 (m, 3H), 7.44 (m, 2H), 7.90 (m, 2H). ^13^C NMR (CDCl_3_): δ 22.0, 70.5, 108.2, 117.8, 122.2, 125.9, 128.5, 128.9, 130.2, 130.6, 138.1, 148.3, 159.0. HRMS (ESI) calcd. for C_17_H_17_N_3_ONa 302.1269 [M + Na]^+^; found 302.1271.

1-(3-isopropoxyphenyl)-4-(2-(trifluoromethyl)phenyl)-1H-1,2,3-triazole. (**9c**) To a solution of **10b** (50 mg, 0.2 mmol) in a MeOH/H_2_O (1:1) mixture, K_2_CO_3_ (23 mg, 0.17 mmol), sodium ascorbate (8 mg,0.04 mmol), CuSO_4_·5H_2_O (1 mg, 0.004 mmol) and **11b** (35 mg, 0.2 mmol) were added. The mixture was stirred overnight, then poured into water and extracted with EtOAc. The combined organic layers were dried with Na_2_SO_4_, filtered and the solvent was removed under reduced pressure. The crude was purified by flash column chromatography on silica gel (*n*-Hex/EtOAc (8:1)) to yield **9c** as a brown solid (43 mg, 73% yield). MP = 90 °C. Rf: EtOAc/*n*-Hex (1:8) = 0.4). ^1^H NMR (CDCl_3_): δ 1.38 (s, 3H), 1.40 (s, 3H), 4.66 (quint, *J* = 6.1 Hz, 1H), 6.97 (dd, *J* = 8.3, 1.8 Hz, 1H), 7.30 (ddd, *J* = 7.9, 2.1, 0.9 Hz, 1H), 7.38 (t, *J* = 2.2 Hz, 1H), 7.42 (t, *J* = 8.2 Hz, 1H), 7.52 (t, *J* = 7.7 Hz, 1H), 7.67 (t, *J* = 7.6 Hz, 1H), 7.79 (d, *J* = 7.9 Hz, 1H), 8.05 (d, J = 7.8 Hz, 1H), 8.18 (s, 1H). ^13^C NMR (CDCl_3_): δ 22.0, 70.5, 108.4, 112.3, 116.2, 121.0 (q, *J_CF_* = 5.6 Hz), 124.2 (q, *J_CF_* = 273.2 Hz), 126.2 (q, *J_CF_* = 5.8 Hz), 127.4 (q, *J_CF_* = 30.5 Hz), 128.5, 129.2 (q, *J_CF_* = 2.2 Hz), 130.6, 131.8, 132.1, 138.0, 144.8, 159.0. HRMS (ESI) calcd. for C_18_H_16_F_3_N_3_ONa 370.1143 [M + Na]^+^; found 370.1140.

### 3.3. Anthelmintic Activity Assays

#### 3.3.1. *C. elegans* Motility Assay

The protocol employed for the automatized L1 motility assay is described in detail in the literature [43]. Briefly, eggs obtained from bleaching were placed on empty magnesium-complemented NGM plates (without bacteria) and were left to hatch for 20 h at 20 °C. L1 worms were collected in a minimum volume of M9 buffer (Na_2_HPO_4_ 6g/L, KH_2_PO_4_ 3 g/L and NaCl 5 g/L). Approximately 250 L1 worms were seeded in each well of a microtiter plate in 80 microliters of M9 buffer with 0.015% bovine serum albumin. Motility was tracked for 30 min in the WMicrotracker device for normalization. After this, compounds of interest in the vehicle (M9 + DMSO 1%) were added in a volume of 20 μL. Compounds were first solubilized in DMSO at a 1 mM compound concentration and then diluted 1:100 in M9 buffer (10 μM compound, 1% DMSO). Motility was tracked for an additional 18 h. Counts per well were normalized with the value obtained previously to compound or control addition. Three replicates were conducted for the 10 µM screening assay and five replicates for each concentration used for the dose-response curves. EC_50_ values were estimated using a four-parameter logistic regression (Hill equation) as implemented in GraphPad Prism 8. The EC_50_ for active compounds are presented with the SD corresponding to the five replicas. The standard deviation for motility at every concentration is automatically generated by the equipment used (Wmicrotracker, Santa Fe, Argentina).

#### 3.3.2. *E. granulosus* Protoscolex Assay

First, 50,000 protoscoleces, obtained from asceptical punction of a single hydatid cyst from a bovine lung, were washed several times with PBS and then incubated at 37 °C, 5% CO_2_, in DMEM supplemented with antibiotics and 20 mM HEPES, pH 6.8. Then, 200 protoscoleces were placed in each well of a microtitration plate and treated with 20 μM of the corresponding inhibitor in 1% DMSO. Protoscoleces were observed under the microscope at an endpoint time of 24 h, and viability was assessed by exclusion of the vital dye eosin at 24 h. The infected bovine viscera were obtained as part of the normal discard processing from a local abattoir. Three replicates were made for each compound for fixed-dose experiments and five for dose–response curves. The EC_50_ for active compounds is presented with the SD corresponding to the five replicas.

### 3.4. In Silico Methods

#### 3.4.1. Homology Modeling

Relevant crystallographic structures of *A. suum* complex II were retrieved from PDB database (PDB ids: 3VR8, 3VR9, 3VRA, 3VRB, 4YSX, 4YSY, 4YSZ, 4YT0, 4YTM, 4YTN, 5C2T, 5C3J). Biological assemblies of these structures were prepared with MOE. Crystal 4YSX was selected as the homology modeling template, due to the similarity of the co-crystallized ligand with Wact-11 family of compounds. *C. elegans* complex II subunits sequences were retrieved from Wormbase [47]. One model was built for each sdha subunit present in *C. elegans*. Several programs and servers (MOE [48], Modeller [49], i-Tasser [50], Swiss-model [51] and Phyre2 [52]) were explored in order to build the homology models. In every case, the default settings of the server/software were used to build the homology model. Visual inspection of the active site, RMSD minimization and quality assessment services (QMEAN [53] and Molprobity [54]) were used to select the best model. MOE was the best performing software in this case. In this software, the 10 best models (scored through the GB/VI scoring function) were retrieved and visually inspected. The best scored model (CA RMSD 0.3, GBVI Score −49,472.7) highly reproduced the quinone-binding site crystallographic geometry and was also the most suitable according to the quality assessment services. The assembled model was reprotonated to simulate a physiological environment with the protonate 3D module of MOE and was minimized using AMBER14-EHT forcefield.

#### 3.4.2. Docking

PDB structures of *A. suum* complex II were used to assess the docking algorithm performance at ligand pose prediction. Autodock 4 [55], AutodockVina [56] and MOE docking module were explored. The only docking procedure that replicated all the ligand poses at RMSD < 1.5 Å was implemented in MOE using Triangle Matcher as placement method (timeout = 3000 s, number of returned poses = 100,000) with London dG as preliminary scoring function. Three hundred poses were generated in the placement stage, which were refined with rigid receptor using GBVI/WSA dG scoring function. In all cases, the pose with the best score was in strong geometrical agreement (RMSD < 1.5 Å) with the corresponding crystallographic structure. 

#### 3.4.3. Membrane Modeling

Each complex II–ligand complex was submitted to the OPM44 database to predict the relative orientation of the complex and the lipid bilayer. CHARMM-GUI [44] membrane builder module was used to simulate and place the mitochondrial inner membrane. POPC, POPE and CL lipids were used in a 2:1:0.2 proportion to build the membrane patch of 90 × 90 Å with the protein centered. All other parameters were those corresponding to the standard CHARMM-GUI protocol.

## 4. Conclusions

Bioisosteric replacement proved to be a fruitful strategy to generate benzamide derivatives. A wide set of compounds was designed, prepared and assayed against *C. elegans* as a nematode model. The best compounds were selected for evaluation against *E. granulosus*. 

The original benzamides were active against L4 and adult worms of *C. elegans* and the parasitic nematodes *Cooperia* spp. and *H. contortus* [8]. These lifecycle stages parasitize the host. In this study, we used L1 since the sensitivity is enhanced [43] and therefore provides more precise information regarding SAR studies. Thioamide **3** and selenoamide **4** presented EC_50_ values in the low μM, comparable to the original benzamide **1a**. It would be interesting to test, in follow-up studies, whether the thioamide and selenoamide bioisosteres may offer a more favorable pharmacokinetic profile. Other bioisosteric replacements turned the compounds to less active or even inactive. The results herein obtained allowed for a structure–activity relationship study, highlighting the importance of the amide group, which seems to be crucial for nematicide activity.

Benzamide **1a** and bioisosteres **3** and **4** were also evaluated against the platyhelminth *E. granulosus*. Benzamide **1a** resulted active with an EC_50_ = 10 μM when examined in protoscolex (literally “protohead”), also referred to as “larval worm”, the infective stage that gives rise to the adult worm inside the definite host. This broadens the scope of target organisms for this family of compounds. Previously reported alignments showed that the quinone (and thus benzamide) binding pocket is highly conserved throughout helminths, both in nematode and plathelminth phyla, explaining this pan-anthelminthic activity for **1a** [57].

Structural analysis of the interaction between benzamides and related compounds and complex II proved to be a useful tool to assist the design of analogs and to provide possible explanations for activity results. The recognition of the potential role of the membrane in the drug binding to a transmembrane enzymatic complex such as complex II could offer new insights in the design of new complex II inhibitors.

## Data Availability

Not applicable.

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
