# Peer review of "Structure-Based Bioisosterism Design, Synthesis, Biological Evaluation and In Silico Studies of Benzamide Analogs as Potential Anthelmintics"

_molecules, 2022, doi:10.3390/molecules27092659_

Round 1

Reviewer 1 Report

In this manuscript, Franco Vairoletti et al. describes the synthesis and assessement of several benzamides as nematicidal agents. The manuscript is well written. However there there are some issues that should be addressed. 

  1. Please provide the RMSD value of docked co-cristalized ligand.
  2. Provide also a 2D interaction diagram in Fig. 5 (it can be generated using Discovery Sutdio software) in order to see in detail the interactions carried out by benzamide 1a. Please also provide a figure for the binding mode of 3 and 4. Finally, discuss the differences among these compunds.
  3. The Homology model protocol should be explained in detail since the authors just mentioned the softwares as well servers that they used.  

Reviewer 2 Report

Comments to the Authors:

This manuscript describes a series of new anthelmintic benzamide analogs synthesized by a method of structure-based bioisosterism. The drug activity and safety of benzamide analogs were evaluated by C. elegans L1 assay and Echinococcus granulosus protoscolex assay. In silico studies, structural analysis of compounds with higher activity was carried out and the potential role of the membrane in drug binding was found, providing a new direction for the design of novel complex II inhibitors. This manuscript can be published in Molecules after the following revisions.

  1. Page 9, line 288, “2.2.1” by exited at the Page 7, line 242, should be changed to “2.2.2” and the author should check it carefully.
  2. Solubility data for the drug should be enclosed herewith.
  3. Why is there no data on other drugs docking in “Docking analysis” section?
  4. Page 1, Introduction, the authors said “However, few options are available; resistance and efficacy problems are common for every anthelmintic family”. Some methods solved resistance and efficacy problems should be mentioned, the following recently published important related paper should be cited: Soc. Rev. 2017, 46, 7021; Chem. Soc. Rev. 2021, 50, 2839; Adv Mater. 2022, 34, 2106388.

Reviewer 3 Report

In manuscript ID molecules-1660620, authors have described the synthesis of a set of new potential anthelmintics structurally related to the benzamide Wact-11 (compound 1a) used as the reference compound. Mainly they have modified the amide bond, replacing it with some bioisosteric moieties. Then they have performed a biological screening on C. elegans, and compounds 3 and 4 emerged as the most interesting derivatives, although little or no improvement was obtained with respect to the reference compound. 
In my opinion this manuscript is not suitable for publication on Molecules, since in its present form the work does not show any progress with respect to the state of the art. However some further investigations, as well as a more concise presentation, could be helpful for potential publication.
1. Since the amide bond is susceptible of hydrolytic metabolism, authors could compare molecules 3 and 4 metabolic stability to investigate if they are endowed with a more favorable pharmacokinetic profile with respect to 1a.
2. The authors declare 15 new compounds, but the manuscript describes the synthesis of 13 new molecules. This should be corrected in the text. Moreover the use of the expression “library of compounds” should be avoided, considering the small number of derivatives.
3. The statistical analysis  must be added to the biological results. How many replicates were performed?
4. The docking study is focused on the binding mode of 1a, which is the reference, but does not analyze the new compounds. Therefore it is not useful to explain the observed results and should be carried out again on the derivatives 3 and 4 in comparison with 1a. Differently, this section should eliminated 
5. English grammar and style need a careful revision

Reviewer 4 Report

The manuscript presents the process of designing, synthesizing and evaluating new compounds using bioisosteric replacement of an amide group present in benzamides (Wact-11 and Wact-12) described previously in the literature.  Eight types of amide replacing groups were selected including ester, thioamide, selenamide, sulfonamide, alkylthio- and oxo-amides, urea, and triazole.

The authors have tested the structure-activity relationship nematocidal activity. The experimental evidence was complemented by in silico structural studies of a C. elegans complex II model as the molecular target of benzamides.

The best activity was observed for thioamide- and selenoamide-derivatives, less active were thiourea- and N-methylthioamide-  derivatives.

The activity thioamide- and selenoamide-derivatives were also tested next to Wact-11 against the flatworm Echinococcus granulosus, and the results proved their anthelmintic potential.

In the introduction, the authors refer to the activities of previously obtained and commercially available compounds (Ref. 8). In Fig. 1 there is some distortion because the structures described in the literature are shown, while the caption relates to the compounds scheduled to be obtained: 1a and 1b.

On page 6 (line 22), Table 1 is incorrectly referred to in the text, it should be Table 3.

Round 2

Reviewer 3 Report

Authors have partially improved their manuscript, and it could be now published  in its present form